# Pre-Clinical Evaluation of Tenofovir and Tenofovir Alafenamide for HIV-1 Pre-Exposure Prophylaxis in Foreskin Tissue

**DOI:** 10.3390/pharmaceutics14061285

**Published:** 2022-06-16

**Authors:** Laura Else, Sujan D. Penchala, Azure-Dee Pillay, Thabiso B. Seiphetlo, Limakatso Lebina, Christian Callebaut, Suks Minhas, Roland Morley, Tina Rashid, Neil Martinson, Julie Fox, Saye Khoo, Carolina Herrera

**Affiliations:** 1Bioanalytical Facility, Molecular and Clinical Pharmacology, Institute of Systems, Molecular and Integrative Biology, University of Liverpool, Liverpool L69 7BE, UK; l.j.else@liverpool.ac.uk (L.E.); sujan-kumar.dilly-penchala@liverpool.ac.uk (S.D.P.); khoo@liv.ac.uk (S.K.); 2Division of Immunology, University of Cape Town, Cape Town 7935, South Africa; tashrico.p@gmail.com (A.-D.P.); bridgette.seiphetlo@gmail.com (T.B.S.); 3Perinatal HIV Research Unit, Faculty of Health Sciences, University of the Witwatersrand, Johannesburg 2000, South Africa; lebinal@phru.co.za (L.L.); martinson@phru.co.za (N.M.); 4Gilead Sciences, Foster City, CA 94404, USA; christian.callebaut@gilead.com; 5Imperial College Healthcare NHS Trust, Charing Cross Hospital, London W6 8RF, UK; suks.minhas@nhs.net (S.M.); rolandmorley@nhs.net (R.M.); tina.rashid1@nhs.net (T.R.); 6Guys and St. Thomas’ NHS Foundation Trust and King’s College London, London SE1 9RT, UK; julie.fox@kcl.ac.uk; 7Department of Infectious Diseases, Faculty of Medicine, Imperial College, London W2 1PG, UK

**Keywords:** HIV-1, foreskin, pre-exposure prophylaxis

## Abstract

Background: HIV-1 pre-exposure prophylaxis (PrEP) has focused predominantly on protective efficacy in receptive sex, with limited research on the dosing requirements for insertive sex. We pre-clinically assessed the ex vivo pharmacokinetic–pharmacodynamic (PK–PD) profile of tenofovir (TFV) and tenofovir alafenamide (TAF) in foreskin tissue. Methods: Inner and outer foreskin explants were exposed to serial dilutions of TFV or TAF prior to addition of HIV-1_BaL_ at a high (HVT) or a low viral titer (LVT). Infection was assessed by measurement of p24 in foreskin culture supernatants. TFV, TAF and TFV–diphosphate (TFV–DP) concentrations were measured in tissues, culture supernatants and dosing and washing solutions. Results: Dose–response curves were obtained for both drugs, with greater potency observed against LVT. Inhibitory equivalency mimicking oral dosing was defined between 1 mg/mL of TFV and 15 µg/mL of TAF against HVT challenge. Concentrations of TFV–DP in foreskin explants were approximately six-fold higher after ex vivo dosing with TAF than with TFV. Statistically significant negative linear correlations were observed between explant levels of TFV or TFV–DP and p24 concentrations following HVT. Conclusions: Pre-clinical evaluation of TAF in foreskin explants revealed greater potency than TFV against penile HIV transmission. Clinical evaluation is underway to support this finding.

## 1. Introduction

Pre-exposure prophylaxis (PrEP) has shown high efficacy against HIV transmission during receptive intercourse [1]. However, the dosing requirements for insertive sex are not known. TAF is a newer version of TDF and, in combination with FTC, has a smaller pill size than FTC-TDF. Whilst TDF-emtricitabine (FTC) has been evaluated in many trials and risk groups, TAF-FTC PrEP has only been evaluated in one PrEP efficacy clinical trial [2,3]. The CHAPS Trial (NCT03986970) aims to optimize on-demand PrEP dosing or both TDF-FTC and TAF-FTC in a pharmacokinetic (PK) and pharmacodynamic (PD) randomized control trial providing oral PrEP prior to voluntary medical male circumcision and evaluating activity using the ex vivo foreskin tissue explant HIV-challenge model [4]. However, no pre-clinical data were available on the anti-HIV activity of TAF in foreskin tissue to help design this comparative clinical trial.

TAF has potential safety advantages over TDF with fewer renal and bone side effects than TDF. It is converted intracellularly to TFV and phosphorylated to the active TFV–diphosphate (DP) form [5]. This accounts for four- to seven-fold higher intracellular concentrations of TFV–DP in PBMC than with TDF/FTC [6,7,8,9]. However, this advantage has not been consistently reported in mucosal tissues. In a single 25 mg dose of TAF study, a large proportion of vaginal and rectal tissue samples had unquantifiable TFV–DP levels, and the metabolite could not be detected beyond 72 h post-dose [10]. TFV–DP concentrations were approximately 1.3- and 13-fold lower in vaginal and rectal tissue compared with concentrations achieved after administration 300 mg TDF/FTC [11]. Similarly, in another pharmacokinetic study, TFV–DP was quantifiable in only 33% cervicovaginal tissue samples after a single dose of tenofovir alafenamide, although TFV–DP concentrations did increase after 14 daily doses [12].

Studies with female genital tract (FGT) models have shown that TAF achieves similar protection against HIV infection at concentrations ~300-fold lower than TFV [13,14]. However, none has been carried out for insertive sex. TFV inhibits ex vivo infection of foreskin tissue explants at 1 mg/mL [15]. However, no studies to date have established a comparison between TFV and TAF in foreskin tissue. Hence, we aimed to define the ex vivo inhibitory concentrations providing pharmacological equivalency between TAF and TFV in foreskin tissue to be used in the CHAPS trial [4].

## 2. Materials and Methods

### 2.1. Reagents and Viral Isolate

TFV and TAF were donated by Gilead Sciences, Inc. (Foster City, CA, USA). A single HIV-1_BaL_ [16] stock (http://www.aidsreagent.org/) (accessed on 19 April 2022) was prepared in activated PBMCs [17].

### 2.2. Human Tissue

Two surgically resected foreskins were collected at Charing Cross Hospital, London, UK, and three at Chris Hani Baragwanath Academic Hospital, Soweto, South Africa. Tissues were collected after receiving signed informed consent from all patients through the Imperial College Healthcare Tissue Bank, approved by Research Ethics Committee Wales (IRAS 17/WA/0161), or through the University of the Witwatersrand Human Research Ethics Committee (180906B) and South African Health Products Regulatory authority (SAHPRA) (20181004).

### 2.3. Cell and Foreskin Tissue Explants Culture

TZM-bl cells [18,19,20] were grown in Dulbecco’s Minimal Essential Medium (DMEM) (Sigma-Aldrich, Inc., St. Louis, MO, USA) containing 10% fetal calf serum (FCS), 2 mM L-glutamine and antibiotics (100 U of penicillin/mL, 100 µg of streptomycin/mL). Cells were tested for mycoplasma contamination and confirmed mycoplasma-free.

Tissues were transported and processed in local sites within 30 min of surgical resection. The tissues were dissected and the outer and inner foreskin specimens were cut into 2–3 mm^3^ explants comprising the inner squamous epithelium, lamina propria, dartos layer and preputial skin, as described previously [15]. Explants were maintained in a non-polarized system in 96-well U-bottom plates with complete high glucose DMEM (containing 10% fetal calf serum, 2 mM l-glutamine, 100 U of penicillin/mL, 100 µg of streptomycin/mL, 80 µg of gentamicin/mL and 2.5 µg of amphotericin B) at 37 °C in an atmosphere containing 5% CO_2_.

### 2.4. Infectivity and Inhibition Assays

The infectivity of the HIV-1_BaL_ stock was estimated in TZM-bl cells (by luciferase quantitation of cell lysates; Promega, Madison, WI, USA) and in activated PBMCs (by measurement of p24 antigen content in cell culture supernatants). The extent of luciferase expression was recorded in relative light units (r.l.u), as described previously [21]. Viral p24 content in the supernatants was measured with HIV-1 p24 ELISA (Innotest HIV antigen ELISA, Fujirebio Europe, Ghent, Belgium), following the manufacturer’s instructions. Viral growth was reported as pg/mL of p24, extrapolated from the p24 kit-supplied standard curve generated by ODs using a sigmoidal dose–response curve (Prism, GraphPad, San Diego, CA, USA). Appropriate dilutions of culture supernatants were applied to ensure that the data were within the 95% interval of the standard OD range.

Inhibition assays were performed using a standardized amount of virus culture supernatant normalized for infectivity. Considering the published data on FGT [14], TZM-bl cells and explants (one outer and one inner foreskin explant were pooled in each well) were exposed in triplicate to serial dilutions of drug at a constant ratio of 1 TAF to 300 TFV. After 1 h at 37 °C, virus was added to TZM-bl cells (10^3.3^ TCID_50_/mL) and left for the time of the experiment. Alternatively, tissue explants were challenged with HIV-1_BaL_ at the HVT (10^4^ TCID_50_/mL) routinely used to obtain productive infection of explants or at a more biologically relevant LVT (10^2^ TCID_50_/mL). Ex vivo dosing with TFV and TAF, along with viral challenge of tissue explants, was performed in a non-polarized manner. After 2 h of incubation, explants were washed with PBS, transferred to fresh plates and cultured for 15 days in the absence of inhibitors. Approximately 50% of the supernatants was harvested every 3 to 4 days and replaced with fresh media. Infectivity was evaluated in supernatants by analysis of p24 concentration (Innotest HIV antigen ELISA).

### 2.5. Bioanalysis

Parallel explant cultures were set up as described for the inhibition assays with the same total incubation times but without viral challenge to measure TAF, TFV and TFV–DP levels in tissues. Quantifications of dosing supernatant (post-3 h incubation), wash solutions (pooled) and 48 h culture supernatant were performed on a SCIEX 5500 triple quadrupole mass spectrometer interfaced with an electrospray ionisation source (AB Sciex Limited, Warrington, UK) operating in SRM and positive ionisation mode. Analyst software (version 1.7) was used for the optimization of tuning parameters and data acquisition. Multiquant (version 3.03) software was used for peak integration and data processing.

Tissue weight (~10 mg) was converted to volume (µL) by dividing by 1.05 g/mL (1.05 mg/µL) and made up to 100 µL with ice-cold methanol and 20 mM EDTA–EGTA (70:30 *v*/*v*). Explants were homogenized in 2 mL tubes (1.4 mm ceramic beads) using a MINILYS homogenizer at 4000 rpm for 120 s. Deuterated internal standards (TFV-d6, TAF-d5 and ^13^C-TFV–DP) were added to all tubes containing 100 µL of homogenate or supernatant prior to solid phase extraction (SPE). Following the addition of 1% formic acid (300 µL), TAF and TFV were extracted using strong cation exchanger SPE cartridges (SOLA SCX (10 mg/mL)). TFV–DP was extracted with a mixture of acetonitrile–formic acid (100 µL; 98:2 *v*/*v*) followed by further sample clean-up using polymeric reverse phase SPE (Strata-X 33 µ (30 mg/1 mL)). TAF and TFV were eluted using a reverse phase Synergi Polar C_18_ column (Phenomenex, Cheshire, UK). The calibration curve ranged between 0.1 and 100 ng/sample (TFV) and 0.05 and 50 ng/sample (TAF). TFV–DP and its internal standard were eluted using a weak anion exchange column (Thermo Biobasic AX, Thermo Fisher Scientific, Waltham, MA, USA). The calibration curve for TFV–DP ranged between 0.07 and 25 ng/sample (sample = 100 µL) on the column.

All analytical methods were validated in accordance with the EMEA guidelines on bioanalytical method validation. The intra- and inter-assay precision (expressed as percent coefficient of variation (%CV)) ranged between 1.91 and 8.62 (TFV) and 3.34 and 9.11 (TFV–DP), and the accuracy (expressed as %Bias) was between −4.08 and 2.83 (TFV) and −4.79 and 5.55, respectively. The average percentage (%) recovery of TFV and TFV–DP from culture media (which was used as a surrogate for a foreskin tissue explant) was 35% and 92%, respectively, and analyte recovery was consistent across the assay calibration range.

### 2.6. Viability Assay

Viability of tissue explants following exposure to drugs was determined by measuring tetrazolium salt [3-(4,5-dimethyl-2-thiazolyl)-2,5-diphenyl-2H-tetrazolium bromide (MTT)] cleavage into a blue product (formazan) by viable cells [22], as described previously [23]. Optical density values obtained with a Synergy-HT (BioTek, Winooski, VT, USA) plate reader were corrected for explant dry weight. Untreated tissue was considered as a positive control for viability (100%). Nonoxynol-9 (N-9) (LKT Laboratories Inc., St Paul, MN, USA) at 2% *v*/*v* was used as a known cytotoxic agent [24].

### 2.7. Statistical Analysis

Drug concentrations were quantified using a ng/sample calibration curve and normalised to ng/gram of tissue, or ng/mL of dosing/wash/culture supernatant for TAF and TFV or to pmol/g or pmol/mL, respectively, for TFV–DP. Values below the assay limit of quantification (<LLQ) were expressed as half the LLQ and subsequently normalised to per gram of tissue. Values below the assay limit of detection (LoD) and with no visible chromatographic peak above the baseline were excluded.

p24 values were calculated from sigmoid curve fitted (Prism, GraphPad) fulfilling the criterion of R^2^ > 0.7. Pearson’s test was used for correlations.

Drug concentrations were log10-transformed and correlated with the corresponding log-transformed p24 level at day 15 of tissue explant culture post-infection for each subject using a Pearson correlation test. *p*-values were determined using a two-tailed unpaired Student’s *t*-test, and a *p*-value < 0.05 was considered statistically significant.

## 3. Results

### 3.1. Ex Vivo PK Equivalency in Foreskin Explants

Tissue explants were dosed with the base compounds TAF or TFV. TFV, instead of TDF, was used to mimic the rapid cleavage of the prodrug to TFV in plasma and kidneys following oral dosing. Ex vivo dosing of foreskin explants to TFV or TAF did not induce cytotoxicity at the concentrations tested in this study (Figure 1). Parent TFV was quantifiable in all foreskin explants after incubation with both TFV and TAF, with approximately five-fold higher levels detected after TAF dosing (Figure 2a, Appendix A). Following incubation with TFV, excess levels of parent drug (~80% of tissue) were present in the culture supernatants, whereas, with TAF dosing, TFV levels in culture media were on average less than 20% of total TFV in explants (Appendix A). TAF was undetectable in tissue explants.

Intracellular TFV–DP concentrations in the explants were approximately six-fold higher after ex vivo dosing with 15 µg/mL of TAF compared with an equivalent dose of TFV (Figure 2b, Appendix A). In both cases, the concentration of TFV–DP achieved in tissue was proportional to dosing (TFV r^2^ = 0.710; TAF r^2^ = 0.998; *p* = 0.017), and there was no evidence of saturation. TFV–DP was unquantifiable in 91% of explants dosed with ≤5 µg/mL (TFV) and 78% dosed with ≤0.15 µg/mL (TAF). TFV and TFV–DP concentrations in tissue were highly correlated following dosing with TFV (r^2^ = 0.7905; *p* < 0.0001) or with TAF (r^2^ = 0.7342; *p* < 0.0001) (Figure 2c,d).

Following incubation with TFV, the summated concentration of TFV–DP measured in the dosing supernatant, culture supernatant and wash solution accounted for 12% (range: 1–29%) of the total amount detected in the tissue explant (Figure 3c,d). TFV–DP concentrations in culture and wash samples after dosing with TAF were below LLD.

### 3.2. Inhibitory Activity in Foreskin Explants

Prior to evaluating the potency of TFV and TAF in tissue explants, we confirmed their activity in TZM-bl cells. As expected, statistically significant greater in vitro activity was observed for TAF (Figure 4a, Table 1), with a 50% inhibitory concentration (IC_50_) three logs lower (IC_50_ = 0.0006 ± 0.0003) than that for TFV (IC_50_ = 0.203 ± 0.069) against HIV-1_BaL_ (*p* = 0.0073). 

Dose–response curves were obtained for both drugs against HIV-1_BaL_ at HVT and LVT in foreskin tissue explants and, as observed in TZM-bl cells, TAF was significantly more potent than TFV (against HVT: TFV IC_50_ = 3.69 ± 0.61 and TAF IC_50_ = 0.018 ± 0.004; *p* < 0.0001) (Figure 4b, Table 1). Equivalency of ex vivo inhibitory potency was established from 3 µg/mL of TAF and 1 mg/mL of TFV using the dose–response curve against the high viral challenge titer (Figure 4c).

Importantly, statistically significant inverse correlations between ex vivo infectivity levels and drug concentrations were observed for both TFV (Figure 5a,b) and TFV–DP (Figure 5e,f) against HVT following ex vivo dosing with TFV (r^2^ = 0.8732; *p* < 0.0001 for TFV; r^2^ = 0.6867; *p* = 0.0001 for TFV–DP) and TAF (r^2^ = 0.8173; *p* < 0.0001 for TFV; r^2^ = 0.6696; *p* = 0.0002 for TFV–DP). However, significant correlations following LVT challenge were observed with ex vivo TFV dosing (r^2^ = 0.6430; *p* = 0.0006 for TFV; r^2^ = 0.5117; *p* = 0.0027 for TFV–DP) (Figure 5c,g) and not TAF (r^2^ = 0.09358; *p* = 0.2675 for TFV; r^2^ = 0.00291; *p* = 0.8486 for TFV–DP) (Figure 5d,h). We further investigated these correlations by assessing whether a non-linear correlation would be a better fit. Analysis revealed that a non-linear fit was possible for TFV levels in tissues following ex vivo dosing with TFV against LVT challenge (Appendix A).

## 4. Discussion

This study showed that higher levels of tenofovir–DP in foreskin explants were obtained following ex vivo dosing with TAF compared to tenofovir. This is in keeping with results obtained from vaginal and rectal tissue [10,11,12]. Our data demonstrated that ex vivo dosing with 15 µg/mL TAF achieved equivalent concentrations of TFV–DP in foreskin tissue (~1000 pmol/gram) to a 1 mg/mL dose of TFV. The highly significant correlations observed between TFV and TFV–DP indicates the physiological relevance of the ex vivo dosing model. The CHAPS trial will confirm the levels of penetration and metabolization of tenofovir and tenofovir alafenamide in foreskin tissue following oral dosing [4]. The ex vivo inhibitory equivalency observed in foreskin explants was the same as that described in FGT models [13,14] and was confirmed by the significant inverse PK–PD correlations.

The lack of ex vivo PK–PD correlation in explants dosed with tenofovir alafenamide and challenged with a low viral titer was due to the limited range of concentrations tested, resulting in an incomplete dose–response curve for tenofovir alafenamide. The four-parameter non-linear fit observed only for the TFV levels in tissue following ex vivo dosing with TFV against ex vivo challenge with LVT could be an artifact due to the limited sample size, and the results from the CHAPS trial will help to clarify this discrepancy.

Both TFV and TAF have been shown to be stable within the time of dosing we used in this study (3 h) in the culture conditions (37 °C in buffered media such as the complete culture media containing DMEM with sodium bicarbonate) [25,26]. Furthermore, we do not expect that TFV in the culture supernatants will have degraded, both during the 48 h post-dosing incubation period and at the time of downstream PK analysis. The culture supernatants were harvested and immediately placed in a −80 °C freezer prior to drug quantification analysis. The samples were analyzed within 11 months from the time of collection/harvesting. We have extensive in-house stability data to suggest that TFV remains stable in plasma for a prolonged period (>2 years) when frozen at this temperature, when subjected to heat treatment (2 × 58 °C; 50 min) and following up to three freeze–thaw cycles. Other research groups have demonstrated TFV (plasma) stability for up to 34 months at −80 °C [27]. Similarly, TFV was shown to be stable (in media) after being left on the bench at ambient temperature for approximately 20 h, and other groups have shown that it is stable in plasma at ambient temperature for up to 144 h [27]. Drug loss into culture media compared to dosing input levels was higher with tenofovir compared to tenofovir alafenamide and can likely be attributed to the presence of either unabsorbed or unconverted drug, as well as loss from the explants by passive diffusion or active cellular efflux. Consistent with previous observations in FGT tissue and urethral secretions [10,28], with the harvesting schedule of this study and despite topical exposure, tenofovir alafenamide was undetectable in foreskin explants, presumably due to the prodrug’s rapid interconversion to tenofovir in tissue. Drug levels in dosing supernatants, wash buffer and culture supernatant samples were expressed as ng/mL of solution, whereas tissue concentrations were derived from a ng/sample calibration curve (i.e., ng per mg of tissue extracted) and subsequently normalized to ng/gram of tissue. We chose to standardize the concentration units for both tissue and solutions based on the assumption that 1 mL is equal to 1 g of tissue, as opposed to using actual values (e.g., ng per mg of tissue explant “on column”), since both tissue weight and volume of the surrounding supernatant/wash solutions harvested from the incubation experiment were variable factors.

The mucosal tissue explant model [29,30,31,32] has limitations, including (i) progressive loss of architecture despite the maintenance of CD4:CD8 T cell ratios and sufficient viability to sustain viral replication for more than 10 days [33]; (ii) a paucity of data regarding the preservation of immune competence [30]; (iii) limitation in demonstrating sterilizing protection; and (iv) an inability to metabolize certain prodrugs, such as TFV disoproxil fumarate, which is the formulated version of TFV for oral administration. Despite these limitations, tissue explants are an important tool for basic research [34,35,36,37,38,39] and pre-clinical screening of PrEP regimens [40] and are increasingly being used in clinical trials for PK–PD assessment [4,41,42,43,44,45,46,47,48,49,50,51,52,53,54,55]. It has been shown that consistent results can be obtained among different laboratories through protocol standardization [56] and that the model can be used to refine animal models and increase their predictive power [57]. Furthermore, in vivo viral replication fitness can be mimicked in mucosal tissue explants [21,57,58] and, following ex vivo challenge, virus has been shown in different mucosal tissue explant models to penetrate the lamina propia with similar kinetics to those observed in vivo [34,59,60].

The primary purpose of this study was to derive suitable and equivalent doses of TAF and TDF to inform the ex vivo PEP dosing in the CHAPS clinical trial and not to specifically define compartmental PK. Thus, the doses used in this study were based on the concentration of TFV that is known to inhibit ex vivo infection of foreskin tissue and not derived from doses administered in vivo. Ex vivo dosing of tissue explants will likely result in higher localized drug concentrations than tissue sampled from orally medicated subjects for a number of reasons, including the ratio of drug-to-tissue surface area, enzymatic activity, incubation time and weight of tissue. Indeed, there is a trend towards lower TFV levels with F–TAF dosing, compared with TDF in cervical/vaginal tissue [10,11,12]. TAF is known to be a substrate of multidrug resistance protein 1 and breast cancer resistance protein, whereas TFV is not. These efflux transporters are expressed within mucosal tissues and may eliminate TAF but not TFV from the localized site in vivo. It is also possible that the inherent activity of such efflux transporters differs in the ex vivo dosing model.

Our study has established ex vivo dosing concentrations to reach PK–PD equivalency between tenofovir alafenamide and tenofovir in foreskin tissue and suggests an improved PK mucosal profile for TAF. Furthermore, it emphasizes the need to assess the pharmacological profile of PrEP candidates in mucosal tissues.

## Figures and Tables

**Figure 1 pharmaceutics-14-01285-f001:**
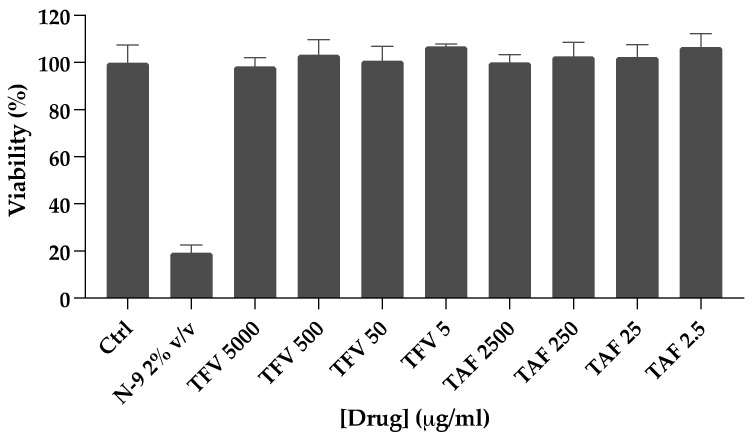
Viability of foreskin explants in the presence of TFV or TAF. Foreskin tissue explants were dosed ex vivo with serial dilutions of TFV or TAF for 24 h. Data are the means (SEMs) from independent experiments performed in duplicate with specimens from two donors.

**Figure 2 pharmaceutics-14-01285-f002:**
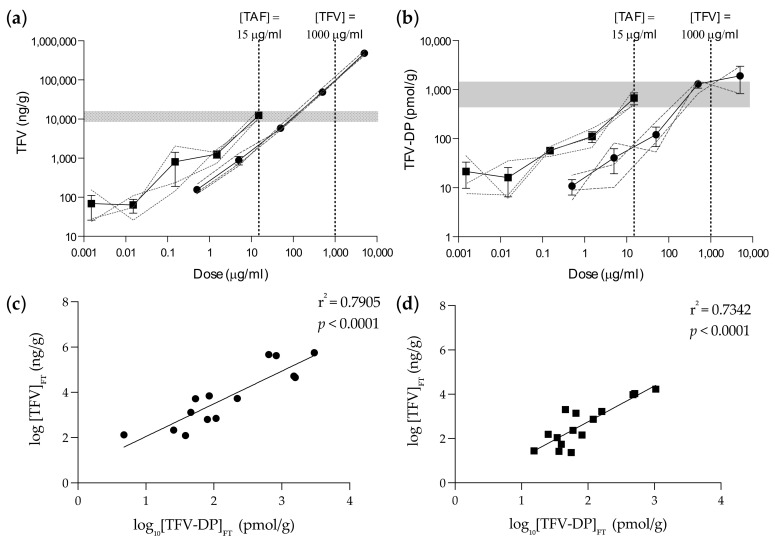
PK analysis. Foreskin tissue explants were dosed with serial dilutions of TFV (●) or TAF (■). TFV (**a**) or TFV–DP (**b**) concentrations were measured in explants 48 h after ex vivo dosing. Dotted lines indicate concentrations of TFV and TAF where equivalence with TFV–DP is observed (grey band in **b**). Dotted grey band (**a**) indicates the level of TFV measured following dosing with TAF. Pearson correlation analysis was performed between explant concentrations of TFV and TFV–DP following dosing with TFV (**c**) or TAF (**d**). Data are the means (SEMs) from independent experiments performed in triplicate with specimens from three donors. FT: Foreskin tissue.

**Figure 3 pharmaceutics-14-01285-f003:**
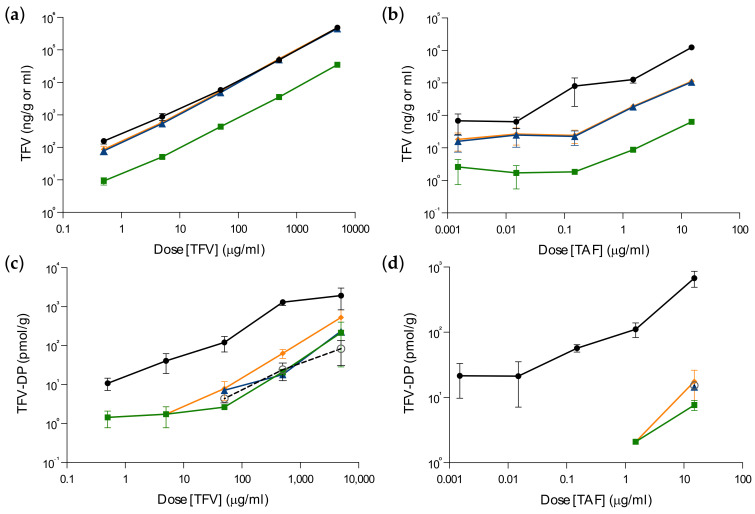
Tenofovir and tenofovir–DP loss following ex vivo dosing of foreskin explants. Tissue explants were dosed with TFV (**a**,**c**) or TAF (**b**,**d**) for 3 h, washed in PBS and then cultured in the absence of drug. TFV (**a**,**b**) and TFV–DP (**c**,**d**) levels were measured in the dosing supernatant after 3 h of culture (◯), in pooled washed buffer (■), in culture supernatant (▲) and in tissue explants (●) harvested after 48 h of culture. Total analyte levels post-dosing (during washing and culture) were calculated (♦). Data are the means (SEMs) from independent experiments performed in triplicate with specimens from three donors.

**Figure 4 pharmaceutics-14-01285-f004:**
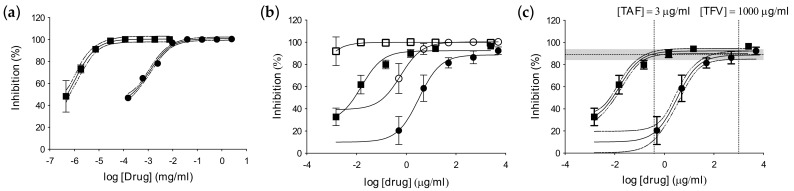
Inhibitory potency of TFV and TAF. (**a**) TZM-bl cells and (**b**,**c**) foreskin explants from HIV-negative donors were dosed with TFV (●, ◯) or TAF (■, □) 1 h prior to challenge with HIV-1_BaL_ at a normalized titer for TZM-bl cells or at a HVT (solid symbols) or LVT (open symbols) for tissue explants. Percentage of inhibition was normalized relative to the r.l.u. or p24 values obtained for cells or explants not exposed to virus (0% infectivity) and for cells or explants infected with virus in the absence of compounds (100% infectivity), respectively. Inhibitory activity equivalency was established in tissue explants (dotted lines and grey band in (**c**)). Data are the means (SEMs) from independent experiments performed in triplicate with specimens from three donors.

**Figure 5 pharmaceutics-14-01285-f005:**
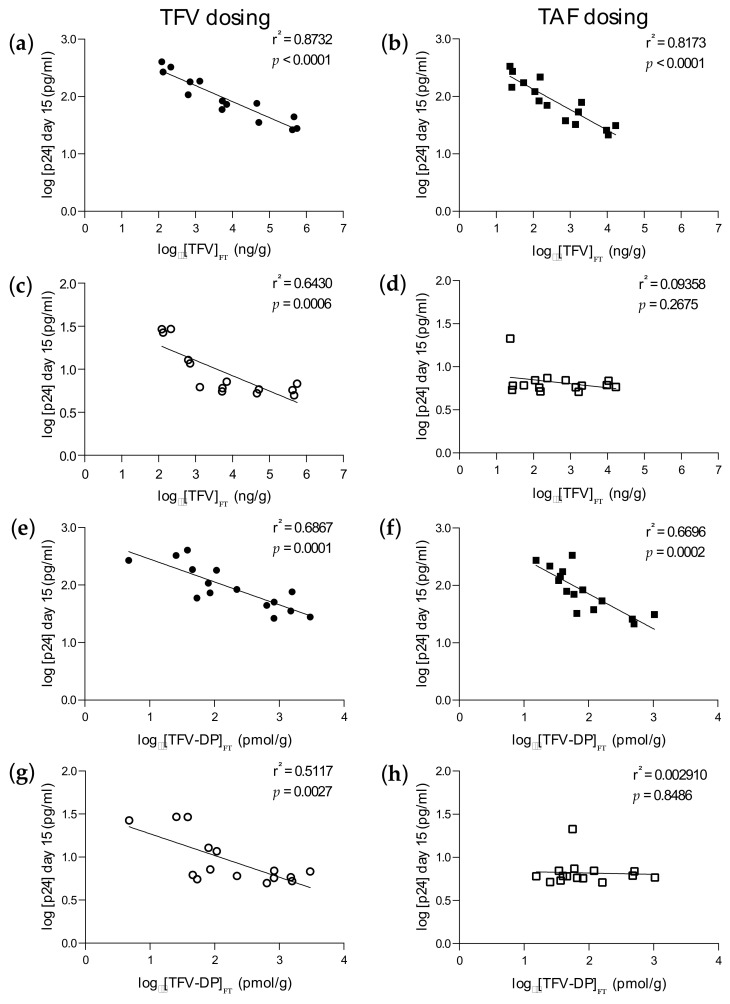
PK–PD correlations in foreskin explants. Pearson correlation analysis was performed between tissue explant concentrations of TFV (**a**–**d**) or TFV–DP (**e**–**h**), and p24 levels in culture supernatants 15 days post-ex vivo challenge with HVT (**a**,**b**,**e**,**f**) or LVT (**c**,**d**,**g**,**h**) following ex vivo dosing with TFV (●, ◯) or TAF (■, □) and challenge with HVT (solid symbols) or LVT (open symbols). Data are the means (SEMs) from three independent experiments performed in triplicate. FT: Foreskin tissue.

**Table 1 pharmaceutics-14-01285-t001:** Inhibitory potency of tenofovir (TFV) and tenofovir alafenamide (TAF) against HIV-1_BaL_.

		TFV		TAF	
Model		IC_50_ (µg/mL)	IC_90_ (µg/mL)	IC_50_ (µg/mL)	IC_90_ (µg/mL)
TZM-bl cells		0.203 (0.069)	5.157 (0.537)	0.0006 (0.0003) **	0.007 (0.002) ****
Foreskin explants	HVT	3.69 (0.61)	435.60 (115.44)	0.018 (0.004) ****	0.90 (0.26) ***
	LVT	N/A	3.17 (0.75)	N/A	0.005 (0.001) *

IC_50_: 50% inhibitory concentration; IC_90_: 90% inhibitory concentration. Data are the means (SEMs) derived from three independent experiments performed in triplicate. HVT: High viral titer; LVT: Low viral titer. N/A: Not applicable, values could not be calculated within the range of concentrations tested. * Statistical significance towards TAF inhibitory concentrations in comparison with TFV inhibitory concentrations was calculated using a two-tailed unpaired Student’s *t*-test (* *p* ≤ 0.05, ** *p* ≤ 0.01, *** *p* ≤ 0.001, **** *p* ≤ 0.01).

## Data Availability

Data available upon request.

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
