# Peer review of "Pre-Clinical Evaluation of Tenofovir and Tenofovir Alafenamide for HIV-1 Pre-Exposure Prophylaxis in Foreskin Tissue"

_pharmaceutics, 2022, doi:10.3390/pharmaceutics14061285_

Round 1

Reviewer 1 Report

In this study, the “Pharmacokinetic and Pharmacodynamic Study of Tenofovir

and Tenofovir Alafenamide for HIV-1 Pre-exposure Prophylaxis in Foreskin Tissue” have been investigated. This study demonstrated the pre-clinical effects of tenofovir (TFV) and tenofovir alafenamide (TAF). The design and method of the study are appropriate. The manuscript is very well prepared. Therefore, I consider the manuscript to be acceptable as it is, without any suggestions or corrections.

Author Response

We thank the reviewer for his positive response to our manuscritp.

Reviewer 2 Report

Line 50-57 please move to discussion, make introduction more focused for aim of the study.

“2.4. Bioanalysis” whole point is not acceptable in current form. Looks the method was not previously published. Please referee range of validation criteria of validation specific regulations in reference and update this section probably with Supplementary file.   

“TZM-bl cells. As expected, greater in vitro activity was observed for TAF” please use numerical data and statistical language is that means statisticaly significant? Unclear in current form

Figure 2. I cant find methodology for this step in current form its unclear

Its unclear what it means “ex vivo dosing” Authors should explain all experiments in Materials and Methods now there is no link between some parts of Results section and Materials and Methods section.

“The highly significant correlations observed between TFV and TFV-DP confirm the physiological relevance of the ex vivo dos- 242 ing model.” Without predictivity analysis is only hypothesis. Please remember that p value and R2 are not a measurements of model predictivity. Please applicate for example leave-one-out method for model validation (observed versus predicted analysis) then you can state it works but current analysis cant proof this concept any way.

“culture supernatants” Authors state that they work with culture supernatants if it means they work with homogenized tissue what was the recovery of the method (please describe as element of method validation).

“r both drugs with greater potency” please remember that homogenized tissue is not model of drug distribution in the living organ especially foreskin tissue

If authors use homogenized tissue should use term spiked tissue not dosed tissue please change the wording

Figure 4 “or LVT (open symbols) for tissue explants. Percentage of inhibition was normalized relative” tissue explants or homogenates.

If the tissue was not homogenized then virus is present only on surface of the tissue (please explain in the papers possible gaps in the model), and interact with drug existing in supernatant   

If the tissue was homogenized then virus was present everywhere in homogenate (please explain in the papers possible gaps in the model) and drug was present everywhere (massive binding for all proteins and damaged fragments)

Unfortunately based on current description the model proposed sound artificial. Please describe model with better way. Explain all pros and cons for the model. Please remember that model not mast be realistic “all models are wrong but some of them are useful” please explain gaps and benefits from the model in context of distribution of the virus and distribution of the drug. Take into consideration organ physiology and drug phys/chem.

Figure 5 please make pairs in one for example a) + b) and show only fit, standard deviation lower for lower fit and SD higher for higher fit.   

  1. c) make appropriate fit it look like log fit or another but not linear please make corrections
  2. e) + f) in one; g) +h) in one

“Drug loss into culture media was higher with tenofovir compared to tenofovir ala- 250 fenamide and likely attributed to the presence of either unabsorbed or unconverted drug, 251 as well as loss from the explant by passive diffusion or active cellular efflux.” Please explain stability of the drug in the culture media and explain in the manuscript then the statement make sense.

Author Response

Please see attached document with responses

Reviewer 3 Report

The manuscript entitled "Pharmacokinetic and Pharmacodynamic study of tenofovir and tenofovir alafenamide for HIV-1 pre-exposure prophylaxis in foreskin tissue" presented by Else et al, show ex vivo characterization of the concentration response of these compounds, in order to determine the best dose to be used in a trial. Overall, the manuscript is scientifically sound and well-written.

The only comments that I have is that the title might be improved, since there is much more to characterize when one studies the PKPD relationship of a drug than the concentration response, and therefore maybe changing those two words in the title might reflect better what the authors are showing.

In addition, the results section could benefit from more numbers included in the text, even though the authors present numerical results in figures and tables, it would make the text more specific. For example, the end of second paragraph of section 3.1 says "TFV and TFV-DP concentrations in tissue were highly correlated" - I would include the numerical r2 value here. Another example would be beginning of 3.2 section, "with a 50% inhibitory concentration three logs lower" - Id show actual results. 

Other than that, I congratulate the team for a very well written paper and well designed experiments. 

Author Response

Please see attached document with responses.

Round 2

Reviewer 2 Report

No additional comments,